# Extended Storage of Beef Steaks Using Thermoforming Vacuum Packaging

**DOI:** 10.3390/foods12152922

**Published:** 2023-07-31

**Authors:** Gabriela M. Bernardez-Morales, Brooks W. Nichols, Savannah L. Douglas, Aeriel D. Belk, Terry D. Brandebourg, Tristan M. Reyes, Jason T. Sawyer

**Affiliations:** 1Department of Animal Sciences, Auburn University, Auburn, AL 36849, USA; gzb0063@auburn.edu (G.M.B.-M.); bwn0004@auburn.edu (B.W.N.); sld0060@auburn.edu (S.L.D.); adb0097@auburn.edu (A.D.B.); tdb0006@auburn.edu (T.D.B.); 2Winpak Ltd., 100 Saulteaux Crescent, Winnipeg, MB R3J 3T3, Canada; tristan.reyes@winpak.com

**Keywords:** color, cook loss, lipid oxidation, storage life, vacuum packaging, warner-bratzler shear force

## Abstract

Extended storage duration often results in negative quality attributes of fresh or frozen beef steaks. This study focused on evaluating the fresh and cooked meat quality of beef steaks stored using vacuum packaging for 63 days. Steaks 2.54 cm thick were packaged into one of three thermoforming films VPA (250 µ nylon/EVOH/enhanced polyethylene coextrusion), VPB (250 µ nylon/EVOH/enhanced polyethylene coextrusion), or VPC (125 µ nylon/EVOH/enhanced/polyethylene coextrusion). Steaks placed in VPA were lighter (L*) and redder (a*) in surface color (*p* < 0.05) as the display period increased, whereas steaks packaged in VPB and VPC became darker. Yellowness, hue angle (Hue°), and chroma (C*) values were greater (*p* < 0.05) in steaks using VPC film as the storage period increased. Calculated spectral values of red to brown were greater (*p* < 0.05) for steaks in VPA and VPB than in VPC. However, steaks placed in VPC films contained greater (*p* < 0.05) forms of metmyoglobin and oxymyoglobin and lower calculated relative values of deoxymyoglobin. In addition, packaging treatment altered (*p* > 0.05) lipid oxidation, but storage time had a greater (*p* < 0.05) influence on purge loss, cook loss, and Warner-Bratzler shear force (WBSF). Current results suggest that the use of vacuum packaging for extended storage of beef steaks (>60) days is plausible.

## 1. Introduction

Packaging is a fundamental part of the food industry that is used to create a product that is not only functional but also convenient for the consumer. Vacuum packaging for fresh meat throughout the various segments of the meat and food industry is increasingly popular in the United States and is under continual innovation [1]. There is a need for centralized packaging methods to increase demand for greater quality and safety of meat cuts for the consumer [2]. Using vacuum packaging requires the placement of beef cuts into plastic bags or pouches and evacuating the atmosphere from within the package. Vacuum packaging can increase the storage life of meat and reduce retail losses, enhance distribution, and maintain meat quality [3].

Thermoforming packaging utilizes heat and pressure to mold a pouch inline using plastic film. After filling the pouch with fresh or cooked meat, a second layer is applied by voiding the atmosphere of the package and sealing with heat. Conventional packaging methods for retail use consisting of polyvinyl chloride (PVC) film and an expanded polystyrene tray have declined in use by almost 46% [1]. Using permeable films for fresh meat, such as PVC, results in greater exposure of the meat surface to detrimental gases, such as oxygen. Plastic films used in thermoforming applications can limit the transmission rate of atmospheric gases to the meat surface and lengthen the stability of surface color on fresh meat products [4].

Meat quality greatly influences the marketability of beef, and research continues to highlight surface color as a factor that consumers continue to use in determining freshness and safety at the time of purchase [3]. Consumers associate and prefer the bright cherry-red color of fresh beef, in contrast to a purplish-red color linked to vacuum-packaged meat as an indicator of wholesomeness [5]. Preferences for a desired red surface color have led to discarding meat that does not meet this parameter and does not guarantee its marketability [6]. It is well known throughout the literature that altering the color of the surface of meat causes profound consumer rejections of the meat at the retail counter [7]. Industry methods have been adopted to minimize this effect, such as controlling the age of fresh beef through packaging and maintaining refrigeration standards that may alter meat characteristics during storage [8]. With advances in technologies, vacuum packaging has caused improvements in the surface color of the meat by maintaining a brighter red surface color for longer periods [4].

Water, as one of the primary components of meat, can be greatly altered by refrigerated storage temperatures, packaging methods, and storage duration [9]. Moisture losses occurring throughout the many phases of meat logistics from farm to consumer have been linked to negative changes in cooking yields, organoleptic properties, and even objective tenderness measurements [10]. Cooking and storing meat can cause tremendous losses of moisture in the meat, ultimately reducing the fragmentation of muscle proteins [10].

Guidelines for cookery and color evaluation highlight a myriad of methodologies for measuring meat quality attributes [11,12]. However, there are no specific guidelines or best practices in storing meat apart from refrigeration for protecting consumer food and meat products [13]. Storing meat products for extended periods has influenced objective tenderness values through moisture loss or degradation of myofibrillar proteins [14,15,16]. Improvements in packaging technologies can enhance the rate at which fresh meat characteristics change during periods of storage prior to consumption.

Therefore, the objective of this study was to evaluate the influence of thermoforming vacuum packaging on the fresh and cooked characteristics of beef steaks after wet aging for 21 days.

## 2. Materials and Methods

### 2.1. Muscle Fabrication

Beef boneless ribeye rolls (Institutional Meat Purchasing Specifications No. 112A) were purchased from a commercial meat processor and transported to the Auburn University Lambert-Powell Meat Laboratory and placed in refrigerated (2 °C ± 1.25 °C) storage (Model LEH0630, Larkin, Stone Mountain, GA, USA). Following 21 days of wet aging, ribeye rolls were removed from their individual vacuum packaging and fabricated. Ribeye rolls (N = 20) were cut to obtain 12 beef steaks 2.54 cm thick using a BIRO bandsaw (Model 334, Biro Manufacturing Company, Marblehead, OH, USA). Steaks from each ribeye roll were allocated randomly to one of three packaging treatments. On days 0, 7, 14, 21, 28, 35, and 42, steaks were removed from the refrigerated display case and measured for instrumental color, lipid oxidation, purge loss, cook loss, and Warner-Bratzler shear force.

### 2.2. Packaging Treatments

After cutting, beef steaks (*n* = 80/treatment) were allowed to bloom to simulate an industry steak cutting application for 30 min at 2 °C (±1.25 °C). After bloom time, each steak was packaged individually into an assigned packaging film using a Variovac Optimus system (OL0924, Variovac, Zarrentin am Schaalsee, Germany). Beef steaks were placed in one of three different thermoforming packaging films—VPA, VPB, or VPC—and sealed with a non-forming layer (NFL) using commercial packaging guidelines (WINPAK, Winnipeg, MB, Canada). Packaging film components, oxygen transmission (OTR), and vapor transmission rates (VPR) of the packaging treatments are presented in Table 1.

### 2.3. Simulated Storage Periods

Initially, packaged steaks were stored frozen in the absence of light at −20 °C (±1.50 °C) for seven days to simulate frozen distribution from manufacturer to retailer at the Auburn University Lambert-Powell Meat Laboratory. Steaks were placed into cardboard boxes, sealed shut, and stored in a blast freezer (Model LHE6950, Larkin, Stone Mountain, GA, USA). After frozen dark storage, steaks were placed into a refrigerated, multi-deck, lighted display case Avantco (Model 178GDC49HCB, Turbo Air Inc., Long Beach, CA, USA) operating at 3.0 °C ± 1.5 °C. Thawed steaks were displayed under constant light for 42 days. The lighting within the retail case consisted of cool LED strips (TOM-600-12-v4-3, Philips Xitanium 40 W–75 W, Korea) with a lighting intensity of 2297 lux (ILT10C, International Light Technologies, Peabody, MA, USA).

### 2.4. Instrumental Color

Instrumental color readings were measured with a HunterLab MiniScan EZ colorimeter, Model 45/0 LAV (Hunter Associates Laboratory Inc., Reston, WV, USA) according to American Meat Science Association (AMSA) Meat Color Measurement Guidelines [17] as described previously by this laboratory [8].

### 2.5. Lipid Oxidation

Steaks were sampled for 2-thiobarbituric acid reactive substances (TBARS) as previously described [4,18], and mg of malonaldehyde/kg of fresh meat was calculated by using the value of 12.21 obtained from a standard curve using a known malonaldehyde solution measured across multiple absorbencies [18].

### 2.6. Purge Loss

Purge loss reduces the weight of a product, is unappealing for many consumers in retail packaging, and ultimately decreases purchase stimulation. Purge loss was collected on fresh (thawed) steaks throughout the refrigerated storage period of 42 days. Steaks were removed from their packaging treatment, blotted dry with a paper towel, and weighed on a balance (Model PB3002-S, Mettler Toledo, Columbus, OH, USA). Purge loss calculations were performed using [(packaged weight − steak weight) ÷ packaged weight × 100].

### 2.7. Cook Loss and Warner-Bratzler Shear Force

Steaks were removed from packages, excess moisture was blotted dry with a paper towel, and then steaks were weighed (initial weight). Steaks were cooked in a convection oven (Vulcan, Baltimore, MD, USA) preheated to 177 °C until the internal temperature of each steak reached 70 °C. Internal steak temperature was monitored with a data logging thermometer (Therma K-Plus, American Fork, UT, USA). Cooked steaks were cooled to room temperature, and final weights were recorded. Cook loss percentages were calculated as follows: [(weight of raw meat samples − post-cook weight of samples) ÷ weight of raw meat samples × 100].

Objective tenderness was measured using Warner-Bratzler shear force (WBSF) with a texture analyzer (Model TA-XT Icon, Texture Technologies Corp., New York, NY, USA). A load cell of 294 N and a crosshead speed of 50 mm/min sheared each core once. Seven cores of 1.27 cm were removed parallel to the muscle fiber from each steak and each core was sheared perpendicular to the fiber direction using previously described methods [11]. The maximum peak force recorded during analysis was reported as Newton (N) of shear force.

### 2.8. Statistical Analysis

The current study was conducted and analyzed as a completely randomized design. Data were analyzed using the GLIMMIX model procedures of SAS (version 9.2; SAS Inst., Cary, NC, USA). Treatment served as the fixed effect, and replication as the lone random effect for meat characteristics instrumental color, lipid oxidation, purge loss, cook loss, and WBSF. Least square means were generated, and significant (α = 0.05) F-values were separated using a pair-wise *t*-test (PDIFF option).

## 3. Results and Discussion

### 3.1. Intrumental Color

Sub-primals in this experiment were wet aged for 21 days before being fabricated into steaks and displayed in multi-deck cases for 42 days. Limited research on extended storage (>60 days) of fresh beef is available in the literature; therefore, this was a novel opportunity to evaluate changes in fresh meat color over long storage periods. Anticipating large changes in myoglobin state in these long-stored meat products, instrumental color readings were used to measure the surface color changes between different pigment forms [19]. There was an interactive impact (*p* < 0.05) of the packaging method and storage period on the fresh surface color (Table 2).

From day 0 to 42, steaks packaged in VPC packaging film were darker (*p* < 0.05) than beef steaks packaged using VPA or VPB (Table 2). Regardless of packaging treatment, L* values initially increased (*p* < 0.05) through day 21. However, as the duration of storage increased, steaks in VPC became darker. Lightness is a characteristic of fresh meat as it blooms, and during lighted display and limited oxygen conditions, oxymyoglobin formation can be altered [20]. An increase in L* using VPA and VPB films is likely the result of film thickness limiting oxygenation of myoglobin and mitochondria resulting in more light scattering on the surface of the steak. Similar changes in lightness were reported in previous studies using vacuum-packaged ground beef over a 14-day simulated display period [20,21]. However, previous literature on the storage of vacuum-packaged whole-muscle cuts after extended wet aging and subsequent fresh storage is limited.

An interaction between packaging film and storage day for objective redness values occurred (Table 2). Steaks were redder (*p* < 0.05) after day 35 of storage when using VPA and VPB consisting of greater barrier properties and a concentration of OMB on the surface of the steaks. However, there were some similarities (*p* > 0.05) among packaging films for redness values from day 0 to 28 of the storage period. Steaks packaged in VPC were less red (*p* < 0.05) and became more yellow (*p* < 0.05) as storage time increased. Similar findings were reported when using vacuum packaging to store foal meat for 14 days in retail display cases [22]. Additionally, the current results tend to agree with others that have evaluated retail color characteristics of vacuum-packaged *longissimus lumborum* and noted an increase in redness (a*) and yellowness (b*) values over retail storage [23].

As expected, hue angle values lacked considerable differences (*p* > 0.05) among all packaging films through the first 21 days of storage (Table 3). However, by day 28 until 42 of the study, steaks packaged in VPC were further (*p* > 0.05) from the true red axis suggesting surface color deterioration was occurring. Surface vividness (C*) was greater (*p* < 0.05) for steaks packaged in VPC than either in VPA or VPB. The changes in vividness suggest that the film thickness in VPA and VPB reduced the rate at which atmospheric gases, such as oxygen, could pass through the film to the surface of the steak and alter the percentage of OMB. It should be noted that current results are similar to previous work on vacuum-packaged beef loins, suggesting that hue angle and vividness stability values deteriorated after peaking during storage [24].

Red-brown ratios (RTB) were calculated from objective measurements of spectral reflectance from 400 to 700 nm. An interaction (*p* < 0.05) for packaging method × day of display is presented in Table 3. Initially (day 0), regardless of the packaging film, RTB values did not differ (*p* > 0.05). However, by day 35, steaks packaged in VPC had a browner surface color (*p* < 0.05). Red-to-brown ratios for beef steaks using VPB packaging film showed a greater color shift (*p* < 0.05) in contrast with the steaks packaged in VPA and VPC films. Furthermore, during the display period, red-to-brown values declined after day 28 (peak) as steaks shifted from a redder to browner surface color. Previous studies have reported similar color shifting of calculated values regardless of packaging method, and it is reasonable that the shift from red to brown is a function of greater metmyoglobin formation throughout the retail display period [4]. It is not surprising that calculated spectral values for instrumental surface color in fresh beef meat are expected to shift from red to brown as the exposure time to atmospheric gases increases. Changes in RTB appear to be related to packaging thickness and the volume of oxygen exposure over time on the surface of the meat.

Calculated relative values of metmyoglobin (MMb) were lower (*p* < 0.05) from day 21 to 42 when using VPA and VPB films (Table 4), whereas steaks packaged in VPC appeared to have a greater (*p* < 0.05) percentage of MMb measured objectively on the surface from day 7 to 42. Calculated relative values suggest the increase in metmyoglobin formation is associated with the oxygen transmission rate that occurred but was not measured throughout the storage period. Previous studies have mentioned that a cause of MMb formation can be accelerated by water loss and heme concentration, but fresh meat in a properly packaged condition should not discolor because of the purge [5].

As the term suggests, deoxymyoglobin can be associated with muscle foods that are not exposed to oxygen, and this myoglobin form can be identified either in vacuum-packaged meat or within the interior of freshly cut meat [24]. Similar to relative MMb values, DMb values in beef steaks packaged using VPA and VPB vacuum-packaged film were greater (*p* < 0.05) than values calculated for steaks in VPC (Table 4). As expected, when using relative values to calculate myoglobin forms of muscle foods, as one form increases (i.e., MMb or OMb) the other forms should decline. Current results tend to agree with results reported on beef steaks using polyvinyl chloride (PVC) overwrap exposed to 35 days in retail display, where the DMb formation was less overall but also increased over time [25].

It is well known that gases, particularly oxygen, from within the atmosphere can react with meat pigments to form a bright red color in contrast to darker purple or brown colors that lack vividness. Calculated OMb values were greater (*p* < 0.05) and declined throughout the entire storage period (Table 4). However, from day 0 to 42, the greatest (*p* < 0.05) decline in relative values of OMb occurred in steaks packaged in VPC. It is quite possible that the changes in relative myoglobin values, especially OMb, are associated with the oxygen transmission rate (OTR) of each packaging film. Oxygenation is a process that occurs when myoglobin is exposed to oxygen and the development of oxymyoglobin causes a cherry-red surface color—this process is commonly referred to as bloom [25].

### 3.2. Lipid Oxidation

Lipid oxidation was measured through the quantification of malonaldehyde (MDA) per kilogram of fresh muscle. There was an interactive effect (*p* < 0.05) of the packaging method × day of display on the lipid oxidation of fresh beef steaks (Table 5). 2-Thiobarbituric acid reactive substance (TBARS) values were greatest for steaks packaged in VPC on day 0 and the least (*p* < 0.05) for steaks packaged using VPB on day 14. It is well known that lipid oxidation values will increase over refrigerated storage periods in fresh and cooked meat products. Current results agree with previous findings, which show the same change in TBARS values in beef cuts aging time during display time using vacuum packaging [26]. Finally, lipid oxidation of fresh steaks using PVC packaging film was reported on days 0 and 14. Surprisingly, for the last retail display day, VPA-packaged film had a higher value in MDA than VPB and VPC films. This contradicts previous studies, which show that greater amounts of oxygen across the packaging material can result in increased catalysis of lipid oxidation [27].

### 3.3. Purge Loss

Measurement of purge loss is commonly reported as a percentage of the meat weight that is lost due to the fluid that is released from the tissue during retail display and is time dependent. An interaction between packaging treatments and storage duration did not occur (*p* > 0.05). Moisture loss was the greatest (*p* < 0.05) on day 42 of the storage period (Table 6). Beef steaks displayed an increasing loss of moisture during storage that may be attributed to the variation in storage temperatures that can occur as a result of display cabinet defrost cycles or operating temperatures. Similar findings were obtained using vacuum packaging methods on the shelf life of chicken where the film thickness had not influenced purge loss when samples were exposed to a stable temperature [27]. Other studies supported that using beef loin cuts to evaluate three packaging methods caused less purge loss when sub-primals were placed in vacuum packages at the end of the storage period [28].

### 3.4. Cook Loss and Warner-Bratzler Shear Force

During cooking, meat can lose a large proportion of its mass, which can be attributed to moisture losses prior to and during the cooking process. There was no interaction between the packaging and storage period (*p* < 0.05) for purge loss, cook loss, or WBSF. As expected, purge loss in packaged steaks increased (*p* < 0.05) with increasing storage time (Table 6), whereas cook loss was greater (*p* < 0.05) in steaks after 28 days of storage (Table 6). These shifts in moisture losses can be caused by the combination of storage time and temperature or cooking conditions, which ultimately can influence the objective tenderness values. Results in the current study agree with previous studies reporting that moisture loss in different retail beef cuts can be altered as storage time increases [29]. Nonetheless, additional aging of meat has shown that lower cook loss can occur in beef cuts aged over 50 days [30].

Changes in moisture during storage and cooking have been well documented to alter meat tenderness. Objective tenderness can be measured via Warner-Bratzler shear force (WBSF) and is often reported in newtons (N) of force. WBSF values were the greatest in steaks on day 21 (*p* < 0.05), but steaks became more tender as storage duration increased (Table 6). Countless studies have concluded that the tenderness and juiciness of meat are affected by heat exposure, and these sensory factors can influence customer satisfaction. Past studies using aging in beef loins around 42 days reported similar trends, where WBSF decreased linearly as the aging period increased [27]. In addition, another study evaluated the tenderness properties of aging beef and concluded that the shear force would decrease when the storage time prior to cooking increased [28].

## 4. Conclusions

Vacuum packaging film thickness does alter the oxygen transmission rate and subsequently influences the fresh characteristics of beef steaks stored for extended periods (>60 days). However, with improvements in vacuum packaging technologies fresh meat can appear redder through objective measurements. As expected, storage duration was a contributing factor that caused differences in purge loss, cook loss, and WBSF, but additional research is needed to further identify the mechanism of these changes. Future studies should be directed towards assessing the organoleptic traits of steaks stored for extended periods by eliciting consumer and trained panelist input on vacuum-packaged fresh beef steaks.

## Figures and Tables

**Table 1 foods-12-02922-t001:** Vacuum packaging specifications for thermoforming films.

Trt. ^3^	Components	OTR ^1^	VPR ^2^
**VPA**	250 µ nylon/EVOH/enhanced polyethylene coextrusion	0.1 cc/sq. m/24 h	2.5 g/sq. m/24 h
**VPB**	250 µ nylon/EVOH/enhanced polyethylene coextrusion	0.1 cc/sq. m/24 h	2.0 g/sq. m/24 h
**VPC**	125 µ nylon/EVOH/enhanced/polyethylene coextrusion	0.6 cc/sq. m/24 h	4.9 g/sq. m/24 h
**NFL ^4^**	110 µ nylon/EVOH/enhanced/polyethylene coextrusion	0.7 cc/sq. m/24 h	6.0 g/sq. m/24 h

^1^ OTR: Oxygen transmission rates. ^2^ VPR: Vapor transmission rates. ^3^ Packaging treatments defined as (VPA, VPB, VPC). ^4^ NFL (Non-forming film).

**Table 2 foods-12-02922-t002:** Interactive impact of packaging method × day on surface color (L*, a*, b*) values during 42 days of refrigerated storage.

	Packaging Treatment ^1^								
Day	Lightness (L*)			Redness (a*)			Yellowness (b*)		
	VPA	VPB	VPC	VPA	VPB	VPC	VPA	VPB	VPC
0	37.76 ^f^	37.94 ^f^	35.72 ^g^	12.23 ^jk^	11.72 ^k^	13.09 ^j^	9.04 ^i^	8.92 ^i^	9.99 ^h^
7	42.25 ^de^	45.53 ^cde^	42.35 ^cde^	20.72 ^i^	21.49 ^hi^	22.38 ^g^	10.82 ^g^	10.95 ^g^	11.90 ^f^
14	44.75 ^ab^	46.29 ^a^	45.10 ^ab^	22.91 ^fg^	22.30 ^gh^	24.16 ^e^	11.67 ^f^	11.39 ^fg^	12.98 ^e^
21	46.23 ^a^	45.78 ^a^	44.79 ^ab^	23.37 ^ef^	23.58 ^ef^	24.20 ^e^	11.64 ^f^	11.83 ^f^	13.80 ^d^
28	41.38 ^bc^	41.03 ^e^	38.44 ^f^	28.15 ^a^	27.91 ^a^	27.52 ^ab^	14.92 ^bc^	15.09 ^b^	17.79 ^a^
35	42.79 ^cd^	42.52 ^cde^	38.34 ^f^	27.33 ^ab^	26.68 ^bc^	25.24 ^d^	14.25 ^d^	14.25 ^cd^	17.77 ^a^
42	43.77 ^bc^	42.97 ^cd^	37.15 ^fg^	26.40 ^c^	25.83 ^cd^	23.52 ^ef^	13.76 ^d^	13.70 ^d^	17.64 ^a^
**SEM**	0.577			0.315			0.243		

^1^ Packaging treatments: VPA (250 µ nylon/EVOH/enhanced polyethylene coextrusion), VPB (250 µ nylon/EVOH/enhanced polyethylene coextrusion), and VPC (125 µ nylon/EVOH/enhanced/polyethylene coextrusion). L* values are a measure of darkness to lightness (larger value indicates a lighter color); a* values are a measure of redness (larger value indicates a redder color); and b* values are a measure of yellowness (larger value indicates a more yellow color). ^a–k^ Mean values within a color measurement lacking common superscripts differ (*p* < 0.05). SEM, Standard error of the mean.

**Table 3 foods-12-02922-t003:** Interactive impact of packaging method × day on calculated spectral values during 42 days of refrigerated storage.

	Packaging Treatment ^1^								
Day	Hue Angle (°)			Chroma (C*)			Red-to-Brown (RTB)		
	VPA	VPB	VPC	VPA	VPB	VPC	VPA	VPB	VPC
0	36.29 ^a^	37.24 ^a^	37.61 ^a^	15.28 ^n^	14.78 ^n^	16.55 ^m^	1.69 ^k^	1.61 ^k^	1.84 ^k^
7	27.56 ^efgh^	26.94 ^efgh^	27.98 ^efg^	23.40 ^l^	24.14 ^kl^	25.36 ^ij^	3.14 ^fghi^	3.34 ^fg^	3.44 ^ef^
14	26.94 ^fgh^	27.05 ^efgh^	28.23 ^def^	25.72 ^hij^	25.05 ^jk^	27.43 ^g^	3.15 ^ghij^	2.97 ^j^	3.22 ^fghi^
21	26.45 ^gh^	26.61 ^gh^	26.67 ^c^	26.12 ^hi^	26.39 ^h^	27.87 ^g^	3.00 ^ij^	3.10 ^ghij^	3.07 ^hij^
28	27.94 ^efg^	28.44 ^de^	32.92 ^c^	31.87 ^ab^	31.74 ^bc^	32.79 ^a^	4.16 ^ab^	4.30 ^a^	3.97 ^bc^
35	27.52 ^efgh^	28.10 ^ef^	35.17 ^b^	30.82 ^cd^	30.25 ^de^	30.88 ^cd^	3.87 ^cd^	3.82 ^cd^	3.26 ^fgh^
42	27.51 ^efgh^	27.91 ^efgh^	36.93 ^a^	29.78 ^ef^	29.24 ^f^	29.43 ^ef^	3.65 ^de^	3.67 ^de^	2.97 ^j^
**SEM**	0.531			0.344			0.087		

^1^ Packaging treatments: VPA (250 µ nylon/EVOH/enhanced polyethylene coextrusion), VPB (250 µ nylon/EVOH/enhanced polyethylene coextrusion), and VPC (125 µ nylon/EVOH/enhanced/polyethylene coextrusion). The hue angle (°) represents the change in color from the true red axis (a larger number indicates a greater shift from red to yellow). C* (Chroma) is a measure of total color (a larger number indicates a more vivid color). RTB is the reflectance ratio of 630 nm ÷ 580 nm and represents a change in the color of red to brown (a larger value indicates a redder color). ^a–n^ Mean values within a color measurement lacking common superscripts differ (*p* < 0.05). SEM, Standard error of the mean.

**Table 4 foods-12-02922-t004:** Calculated spectral values for the interactive impact of packaging method × storage day.

	Packaging Treatment ^1^								
Day	Metmyoglobin (MMb)			Deoxymyoglobin (DMb)			Oxymyoglobin (OMb)		
	VPA	VPB	VPC	VPA	VPB	VPC	VPA	VPB	VPC
0	41.51 ^b^	43.60 ^a^	41.32 ^b^	8.97 ^jk^	9.03 ^k^	9.20 ^j^	49.52 ^b^	47.37 ^c^	49.48 ^b^
7	15.95 ^j^	15.29 ^j^	17.30 ^hij^	32.12 ^i^	33.45 ^hi^	33.44 ^fg^	51.93 ^a^	51.27 ^a^	49.27 ^b^
14	18.08 ^hi^	19.33 ^fgh^	21.08 ^fg^	35.67 ^fg^	34.25 ^gh^	34.74 ^def^	46.25 ^c^	46.41 ^c^	44.18 ^d^
21	21.21 ^f^	20.92 ^fg^	25.46 ^e^	36.27 ^ef^	36.06 ^ef^	32.53 ^g^	42.52 ^def^	43.02 ^de^	42.01 ^ef^
28	16.58 ^ij^	15.96 ^j^	26.29 ^e^	43.09 ^a^	43.10 ^a^	35.92 ^d^	39.33 ^ghi^	40.94 ^fg^	37.79 ^i^
35	18.56 ^hi^	19.10 ^gh^	32.38 ^d^	43.03 ^ab^	41.28 ^bc^	29.73 ^h^	37.96 ^hi^	39.61 ^hg^	37.90 ^i^
42	19.30 ^fgh^	20.81 ^fg^	36.78 ^c^	41.40 ^c^	39.10 ^cd^	25.26 ^i^	39.30 ^ghi^	40.08 ^g^	37.96 ^hi^
**SEM**	1.062			0.577			0.613		

^1^ Packaging treatments: VPA (250 µ nylon/EVOH/enhanced polyethylene coextrusion), VPB (250 µ nylon/EVOH/enhanced polyethylene coextrusion), and VPC (125 µ nylon/EVOH/enhanced/polyethylene coextrusion). Relative values of metmyoglobin (MMb), deoxymyoglobin (DMb), and oxymyoglobin (OMb) using spectral values. ^a–k^ Mean values within a color measurement lacking common superscripts differ (*p* < 0.05). SEM, Standard error of the mean.

**Table 5 foods-12-02922-t005:** Interactive impact of packaging method × day of display of TBARS on beef steaks during 42 days of refrigerated storage.

Storage Day								
	0	7	14	21	28	35	42	SEM
**VPA ^2^**	0.84 ^de^	0.85 ^de^	0.91 ^bcde^	0.86 ^de^	0.92 ^bcde^	0.85 ^de^	0.93 ^bcde^	0.103
**VPB**	0.91 ^bcde^	0.90 ^cde^	0.80 ^e^	1.03 ^b^	0.95 ^bcd^	0.89 ^cde^	0.88 ^cde^	0.106
**VPC**	1.16 ^a^	0.88 ^cde^	1.01 ^bc^	0.96 ^bcd^	0.98 ^bc^	0.84 ^de^	0.88 ^cde^	0.102

BARS: 2-thiobarbituric acid reactive substances are reported as mg/kg of malonaldehyde in fresh tissue. A larger value is indicative of greater oxidation. ^2^ Packaging treatments: VPA (250 µ nylon/EVOH/enhanced polyethylene coextrusion), VPB (250 µ nylon/EVOH/enhanced polyethylene coextrusion), and VPC (125 µ nylon/EVOH/enhanced/polyethylene coextrusion). ^a–e^ Mean values lacking a common superscript differ (*p* < 0.05). SEM, Standard error of the mean.

**Table 6 foods-12-02922-t006:** Effect of storage day on purge loss, cook loss, and Warner-Bratzler shear force (WBSF) of beef steaks during 42 days of refrigerated storage.

Storage Day								
	0	7	14	21	28	35	42	SEM
**PL (%)**	7.05 ^bc^	6.41 ^c^	7.03 ^bc^	8.08 ^a^	7.50 ^ab^	8.24 ^a^	8.28 ^a^	0.285
**CL (%)**	25.79 ^bcd^	22.84 ^d^	23.24 ^cd^	24.50 ^bcd^	35.09 ^a^	30.03 ^abc^	30.05 ^ab^	2.315
**WBSF (N)**	17.17 ^abc^	18.25 ^ab^	18.03 ^abc^	19.12 ^a^	13.77 ^d^	15.79 ^bcd^	15.33 ^cd^	0.939

Purge loss is expressed in percentage (PL), cook loss is expressed in percentage (CL%), and Warner-Bratzler shear force is reported in Newton (WBSF (N)). ^a–d^ Mean values within a row lacking common superscripts differ (*p* < 0.05). SEM, Standard error of the mean.

## Data Availability

Data presented in the study are available upon request to the corresponding author.

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
