# Peer review of "Extended Storage of Beef Steaks Using Thermoforming Vacuum Packaging"

_foods, 2023, doi:10.3390/foods12152922_

Round 1
Reviewer 1 Report
The paper "Extended storage of beef steaks using thermoforming vacuum packaging" covers an interesting topic in the animal food technology.
In the Materials and Methods the Authors report that they have also carried out simulated storage periods for 7 days at -20 C (Section 2.3). Are the results referred only to purge loss? I suggest to declare in the Materials and Methods.
Section 2.7. Why do the Authors sheared meat parallel to the fiber direction and not perpendicular, as AMSA 1995 (Research Guidelines for Cookery) suggests?
Results and Discussion. The Tables 2 and 3 in my opinion are not easy to read. I suggest to change format or to explain in footnote how read the comparisons.
Finally, there is some typing error, as Rows 46-48 (may meat characteristics) or row 165 (weigh).
Author Response
Please find attached a completed list of responses to each review comment listed below. We appreciate all of the effort in reviewing our submission and believe that the changes made have improved the readership of our work.

Reviewer 2 Report
The manuscript Foods-2531166 entitled " Extended Storage of Beef Steaks Using Thermoforming Vacuum Packaging”
Overall, the paper is well written and very well explained.
I have few observations that need to be addressed to improve the paper’s quality.
Introduction
The introduction section does not explain what was done on topic analyzed. Authors spend too many spaces in general information about irrelevant topics. Authors must introduce thermoforming vacuum packaging in the introduction and differentiate this with the conventional vacuum packaging techniques. Additionally, need to discuss merits and demerits of this packaging with reference to the published data. I love to see the cost benefit analysis of this packaging (briefly). More focus should be on this packaging type rather spending too much time in general meat quality.
2.1 Muscle Fabrication
Authors must include animal related information such as breed, live weight, age, carcass weight. As these parameters are directly linked with the ultimate meat quality.
Line 83: Please provide chiller specifications here. It is very important. It is lab type chiller or industrial type chiller? As temperature is very important to determine the meat quality.
Line 88: Please mention the number of steaks used for evaluation on each sampling day?
Q. Why pH has not been measured? For any type of experiment focusing color chemistry pH is very important indicator that directs final color of meat. If data is available, please include. How the authors ruled out the incidence of DFD?
Line 94: Please provide chiller specifications here.
Line 94: Why there is a need of blooming at this stage? Because in this way, the final color intensity at retail shop will be reduced due to already sufficient conversion of OxyMb to MetMb in the absence of air inside vacuum packs.
2.4 Instrumental Color
No need to provide complete details of methodology for already well-established techniques. Just provide suitable reference and make it short.
2.5 Lipid Oxidation
No need to provide complete methodology. It is well-established method. Reference is enough. Delete lines 136-146
Author Response
Please find attached a copy of the responses completed and truly appreciate the time and effort to review our submission. We believe that the comments incorporated have improved the readership of our submission.

Round 2
Reviewer 2 Report
I am satisfied with authors response.